# An Evolution of Reporting: Identifying the Missing Link

**DOI:** 10.3390/diagnostics12071761

**Published:** 2022-07-21

**Authors:** Sara Harsini, Salar Tofighi, Liesl Eibschutz, Brian Quinn, Ali Gholamrezanezhad

**Affiliations:** 1British Columbia Cancer Research Center Vancouver, Vancouver, BC V5Z 1L3, Canada; sara.harsini@gmail.com; 2Department of Radiology, Keck School of Medicine, University of Southern California (USC), Los Angeles, CA 90007, USA; salar.tofighi@usc.edu (S.T.); liesl.eibschutz@med.usc.edu (L.E.); bqmemd@gmail.com (B.Q.)

**Keywords:** interprofessional communication, interprofessional collaboration, patient safety, closed-loop reporting, artificial intelligence

## Abstract

In recent years, radiologic imaging has undergone tremendous technological advances and is now a pillar of diagnostic and treatment algorithms in clinical medicine. The increased complexity and volume of medical imaging has led clinicians to become ever more reliant on radiologists to both identify and interpret patient studies. A radiologist’s report provides key insights into a patient’s immediate state of health, information that is vital when choosing the most appropriate next steps in management. As errors in imaging interpretation or miscommunication of results can greatly impair patient care, identifying common error sources is vital to minimizing their occurrence. Although mistakes in medical imaging are practically inevitable, changes to the delivery of imaging reporting and the addition of artificial intelligence algorithms to analyze clinicians’ communication skills can minimize the impact of these errors, keep up with the continuously evolving landscape of medical imaging, and ultimately close the communication gap.

## 1. Introduction

Current studies estimate that errors are inherent in approximately 4% of radiologic interpretations rendered by radiologists in their daily practice [1]. In addition, between 44,000 and 98,000 Americans die each year as a result of medical errors [2]. Because medical imaging plays a considerable role in obtaining the correct diagnosis, it is reasonable to conclude that the high prevalence of diagnostic unreliability in medical practice is partly attributable to the errors of radiologists [3]. As many of these errors can cause harm to patients, it is imperative to identify the source of these mistakes, as well as to discuss ways to enhance the delivery of imaging reporting. In addition, it is essential that we consider other potential strategies to minimize error rates, such as the use of artificial intelligence algorithms to analyze clinician’s communication skills, in order to improve communication between radiologists, patients, and other key players in the multidisciplinary team. Ultimately, by elucidating the most effective means of mitigating errors in the practice of radiology and identifying supplemental strategies to catch systematically missed errors and improve communication, patient outcomes will undoubtedly improve.

## 2. Errors: A Cost Analysis

In order to decrease the number of errors that occur within radiographic imaging, it is imperative to first define what an error is. For instance, if two radiologists disagree over a finding, is it safe to assume that one of them is making an error [4]? The definition of an error that we will utilize will be in line with Onder et al., who stated that an error is “no uncertainty about the correct finding, with no possibility for dispute or disagreement” [5]. Thus, a justifiable difference in opinion would not constitute a true error but would, rather, be a discrepancy.

Legitimate medical errors carry both a financial and a legal cost. Throughout the United States, the annual cost of these mistakes is roughly USD 37.6 billion, with USD 17 billion of this derived from preventable errors [2]. In addition, a paper by Guardado reviewed the claims frequency data from the American Medical Association’s (AMA) 2016 Physician Practice Benchmark Survey and found that being sued is fairly common for physicians [6]. A total of 34% of all physicians were found to have been sued once, and 16.8% have been sued two or more times [6]. In addition, 68 liability claims were filed per every 100 physicians, on average. However, there were significant differences in claims frequency among specialties. Approximately 37% of surveyed radiologists had been sued at least once. Other studies reported similar findings, with an investigation by the Physician Insurers Association of America revealing that radiologists rank 6th among physicians in the number of malpractice suits in which physicians have been named as defendants [7], even though radiologists constitute merely 8.0% of the U.S. physician population [8]. However, most of these detected errors are minor and do not result in patient harm. Those that are serious in nature are usually promptly identified and corrected.

Poor communication also carries a significant financial burden for healthcare organizations. One study noted that communication failures comprise 23% of radiology malpractice claims [9]. Although errors in communication may be more amenable to change compared to other sources of error, these communication failures are highly costly; a U.S. National Library of Medicine study determined that communication breakdowns that increase patient length of stay cost U.S. hospitals over USD 12 billion annually [10].

## 3. How We Report Findings

Radiologists play an essential role in diagnostic decision-making and treatment, as they are usually the first to know about critical imaging findings that warrant immediate intervention. After identification, it is customary for the radiologist to prioritize communication of these findings to the ordering physician and document the conversation. For many uncomplicated cases, this is standard procedure and issues in reaching the correct diagnosis do not arise. However, in patients with non-typical clinical pictures, complex medical-surgical history, or discrete/no imaging findings, arriving at the correct diagnosis can pose quite a dilemma for both the radiologist and the ordering provider. Additionally, the clinician’s differential diagnosis may change after receiving updated labs or imaging and with further probing of the history and physical exam. Thus, if there is a communication breakdown at this stage, the change in the differential can easily evade the radiologist. Inconsistent communication between radiologists, primary care physicians, sub-specialists, and emergency department providers can also yield non-cooperation at this stage, resulting in a delay in diagnosis and associated decreased available treatment options. Now, with rising volumes of imaging studies and increased use of remote readers/teleradiology, there is greater potential for these scenarios to occur and increased opportunity to harm patients.

Ultimately, communication errors can occur at any step of the process, from ordering the study to interpretation, or discussion with the referring physician. However, a study by Siewert et al. found that over 50% of these errors occur at steps other than communication of results [11]. Instead, most gaps in communication occur during scheduling, ordering, conducting, or interpreting the study. These authors also noted that among the 380 communication errors found within a radiology department, 37.9% affected patient care, with over 85% of these errors having a moderate or major impact (i.e., unnecessary imaging or surgical intervention) [11]. Issues can also arise within the emergency department, where patients are often discharged prior to creation of the finalized report. This can engender a communication breakdown as, frequently, new or changed test results are not communicated to the patient and important information is often not relayed between hospital-based physicians and primary care physicians. In fact, certain studies have reported that direct communication between primary care physicians and hospitalists only occur in 3–20% of discharges [12]. Thus, it is imperative to learn from successful clinician interactions and identify the most common communication gap scenarios to ultimately improve physician collaboration. In addition, clinicians globally should implement closed-loop communication, a process that consists of not only receiving information, but also confirming/cross-checking this information for accuracy [13]. In essence, it is a form of fact-checking so no information can slip through the gaps.

## 4. BI-RADS: An Example of Successful Image Reporting

Breast imaging has successfully pioneered the standardization of image reporting with the breast imaging reporting and data system (BI-RADS) and closed-loop, multidisciplinary reporting, with breast tissue sampling. In this capacity, the radiologist performing tissue sampling plays a key role in collaboration with the surgical pathologists and surgeons to confirm the accuracy of the diagnostic interpretation. However, the breast imaging field has not always excelled in communication and reporting. The evolution of the reporting system can be attributed to the development of image-guided breast biopsies, a minimally invasive method used to sample nonpalpable mammographic lesions considered sufficiently suspicious for carcinoma [14]. This technological advance prompted a discussion by a national task force comprised of representatives from the American College of Surgeons, the American College of Radiology, and the College of American Pathologists, to assess the issues surrounding image-guided breast biopsies. Subsequently, a report on these issues was published in 1997, which stands as the standard guidelines for this technique to this day [15]. In brief, the results of the biopsy should be correlated with all available patient data, including clinical observations, imaging, and pathologic findings, in order to validate the diagnosis and choose a treatment path. In addition, this task force declared that upon availability of the histopathology report, it would be the duty of the radiologist who performed the biopsy to evaluate the imaging workup and decide whether the histopathology findings are concordant or discordant with the imaging characteristics [15]. In case of radiology/pathology discordance, two hypotheses must be entertained:

First, it can be concluded that a representative sample of the lesion was not obtained, and that a repeat biopsy is indicated; second, the original diagnosis was incorrect, and an imaging re-review is necessary. In a study that insisted on cooperation between specialists involved in a patient’s breast biopsy, the authors discovered that approximately 47% of repeat biopsies in the setting of discordance between imaging and pathology resulted in a final diagnosis of carcinoma [16]. Ultimately, the implementation of a closed-loop system of communication by the joint task force led to higher diagnostic rates of patients with breast carcinomas and increased rate of timely diagnosis.

## 5. The Missing Link in the Chain of Reporting: Let’s Close the Loop

In order to generate this medium for discussion and collaboration, we suggest adding an extra component to radiology reports, which will need to be filled in by the referring clinician. This component will characterize the report’s concordance or discordance with the patient’s clinical presentation. This strategy ensures that the ordering clinician is identifiable on all imaging reports and that the responsible clinician has considered the information provided by the imaging report in the context of the workup at that point. Categorizing the radiology report as either “Concordant” or “Discordant” with respect to the clinical picture also places equal responsibility on the radiologist and ordering physician. In the event of “Discordant” imaging with respect to the clinical picture, the ordering physician would be obligated to initiate a conversation with the radiologist. This documented conversation will provide the radiologist with updates related to the patient’s clinical picture and workup, which may clarify the etiology of imaging findings, lead to alternative interpretations of the imaging, or show the necessity of additional workup or specialty referrals. Implementation of this process will be incentivized by the involvement of the covering insurance providers, as the insurers’ payment to the referring physician will be contingent upon completion of this component in the radiology report, as well as appropriate documentation of the communication. Figure 1 graphically illustrates the proposed closed-loop communication system.

However, even with this closed-loop system of communication implemented and the addition of an extra concordant/discordant component, issues can still arise. In a study by Reda et al., the authors reported that over 20% of clinicians did not read the radiology report for the patient in question [17]. Other authors described similar findings, with one study stating that only 55.7% of referring clinicians read the radiologic report in full, while the rest skim, or do not look at the report at all [18]. In addition, many structural problems inherent in the hospital system can arise and yield barriers in communication. For instance, incoming calls from referring physicians are often misdirected to the incorrect radiology workstation. A study by Voutsinas et al. reported that roughly 74% of radiology residents receive misdirected phone calls at least twice a day [19]. In addition, a radiologist that was on call during a previous shift may not be on call when the film is sent back, thereby further delaying diagnosis and treatment. Thus, while direct involvement of the referring physicians can serve as a final quality assurance check and improve patient safety, many procedural issues are still highly prevalent and need to be mitigated.

Even in situations where a discordance between the ordering clinician and reading radiologist has not been identified, many authors discuss the value of returning the pathology report to the original radiologist to re-read, once more thorough patient information has been attained [20]. This, ultimately, results in more accurate diagnoses and improved patient outcomes. In addition, this technique has been shown to lower the number of malpractice litigations and decrease stress and costs to both the medical system and providers [21]. However, most institutions do not require referring physicians to communicate with the interpreting radiologist about results in the context of a patient’s clinical picture. Nor are there escalation procedures in cases of communication breakdown. In addition, re-reading films is often unrealistic, due to the high workload of radiologists and structural issues within the healthcare system [18].

These limitations ultimately emphasize that further strategies may need to be implemented beyond the addition of a concordant/discordant categorization system. One potential idea was to create a structured reporting template, to ensure that communication between the radiologist and the referring clinician took place. These templates included three elements: (when) when the communication occurred, (how) by what method the communication took place, and (who) to whom the findings were communicated [22]. When tested in the clinical setting, the authors reported that communication of all three required items (when, how, who) was achieved in almost 100% of the reports containing the template, versus 5% of reports without the template. Strategies like these ultimately enforce interprofessional communication and guarantee that critical findings are communicated. In a study by Wright et al., the authors discussed other potential interventions to fix this communication breakdown, including having more than one person responsible for test results, automatically sharing results with primary care providers, and having scheduled follow-up times placed in the schedule [9]. Although successful in theory, consideration of both clinician time and resources must occur in order to successfully implement these novel strategies. Ultimately, the American College of Radiology defines an effective mode of communication as a method that satisfies the need for timeliness, encourages physician communication, and minimizes the risk of communication error [23].

## 6. Artificial Intelligence: A Potential Solution

Interestingly, certain authors have begun to discuss a potential niche for artificial intelligence (AI) in analyzing clinician’s communication skills and providing feedback [24]. Although communication skills are heavily taught in medical school, the skills often remain basic and decline once schooling has been completed. However, by having their communication skills routinely assessed via AI, clinicians may constantly improve and receive personalized feedback regarding their performance. In this capacity, AI algorithms can evaluate three separate competencies: first, AI can assess the meaning of words, and analyze how well clinicians and patients understand one another; second, AI can identify negative communication patterns, such as interrupting the other person, and simultaneous chatter; third, machine learning analytical methods can examine pitch, pace, and style in communication. This, in turn, could identify if certain tones of voice or paces of speech can influence patient understanding or motivation. In an article by Ambady et al., the authors discovered that a correlation existed between a surgeon’s tone of voice and the likelihood of facing a malpractice suit [25]. Thus, improving subtle conversation skills can ultimately minimize the number of malpractice litigations, as well as improve patient adherence and outcomes.

However, limitations to these techniques may arise when considering the complexity of human nonverbal communication. For example, Butow et al. notes that using the 43 muscles of our face, we can generate over 10,000 unique facial expressions at a single time point [26]. Thus, in situations where AI techniques are not actively looking at the patient, only analyzing speech may not be enough to effectively characterize the conversation. Ultimately, further research is necessary to help AI methodologies recognize these nonverbal cues in real-time and generate computational algorithms that can pick up the subtlety of human interactions.

In addition to utilizing AI algorithms to facilitate more effective patient–clinician conversations, these techniques are also being investigated in improving communication between radiologists and other key players in the multidisciplinary team. As we have stated, the current communication model between radiologists and the ordering physician is not optimal, particularly in patients that need to be monitored over time. Thus, AI techniques can engender a gold standard for radiology reporting by generating standardized terminology and outlining key objectives that encompass a “good report” [27]. In addition, certain authors note that AI methodologies could potentially shift text-based reports to an interactive, quantitative type, thus improving collaboration and understanding within the team [28]. Thus, the potential role of artificial intelligence techniques in improving communication within the healthcare team are endless.

Ultimately, AI can play a vital role in identifying cases where the radiologic findings are discordant with the clinical picture and in detecting individuals with critical findings, yet no documentation of proper communication between clinicians. Although this will require future research and more advanced AI training, it could be a crucial step in closing the loop in the chain of reporting.

## 7. Conclusions

Until these novel artificial intelligence techniques are fully integrated into the clinical realm, the commencement of required conversations should be implemented if a discrepancy arises between the ordering physician’s views of the patient’s presentation and the radiologist’s report. Thus, radiologists and ordering physicians should reconvene at the completion of the initial diagnostic workup, to determine whether or not the completed imaging is satisfactory or if additional testing is indicated. Although identifying the physician responsible for imaging follow-up is a complex problem in itself, these conversations are critical, to ensure the efficacy of image reporting and optimal patient care ^iii^. Collaboration between the ordering physician and radiologist will lead to a more efficient diagnostic workup and more informed imaging interpretation.

In addition, with the advent of the recommended closed-loop system, radiology departments can collaborate with referring physicians more effectively, preventing missed diagnoses, and consequently decreasing the medical-legal liability associated with these mistakes. Although artificial intelligence techniques may be useful in analyzing clinicians’ communication skills, further research is necessary before this is fully incorporated into the field of radiologic imaging. Thus, it is imperative to focus on modifying the current method of radiology reporting, in order to close the communication gap between radiologists and ordering physicians. The concepts outlined in this article could stand as an accelerator from which these critical discussions are triggered. Therefore, we call upon key stakeholders, incorporating physicians, policymakers, professional societies, and malpractice insurers, to engage in this conversation, in order to ensure radiology reporting evolves with its rapidly changing environment.

## Figures and Tables

**Figure 1 diagnostics-12-01761-f001:**
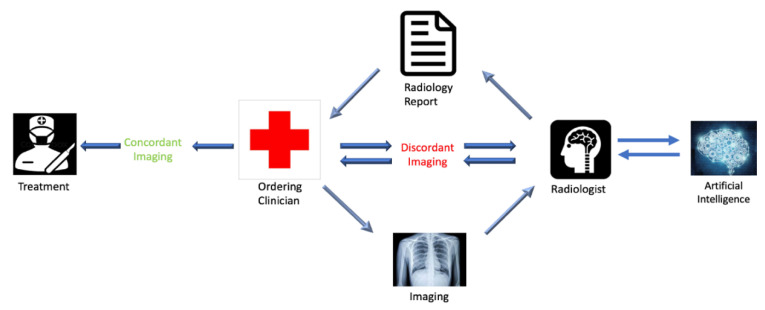
Schematic flow chart showing the proposed closed-loop system of communication. The notable difference between the proposed system and the current system is the required communication between the ordering physician and the radiologist in the event of a discordant imaging report, differing to patient presentation, as determined by the ordering physician. It also includes the potential addition of artificial intelligence which will be discussed in Section 6.

## Data Availability

Not applicable.

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
