# Peer review of "An Evolution of Reporting: Identifying the Missing Link"

_diagnostics, 2022, doi:10.3390/diagnostics12071761_

Round 1
Reviewer 1 Report
Assistant Editor,
Diagnostics, MDPI
Dear Mr. Radu Danila;
This manuscript provided interesting information regarding “errors” on imaging reports. To decrease “error”, the authors recommend “closed-loop system of communication” and also described usefulness of artificial intelligence as a potential solution. I believe that this manuscript would provide special interest to journal readers. Therefore, this manuscript should be accepted for publication. However, I consider some problematic points in this manuscript, as follows.
Major points: I think that this manuscript seems to be relatively lengthy. The authors described “Discussion” separately, but discussion is also included in other sections. I believe that the authors should shorten the manuscript, in particular the length of “4. The Missing Link in the Chain of Reporting: Let’s Close the Loop”
Minor points:
Refence number: 1, 2, 3, … not i, ii, iii, ..; in addition within parenthesis, not right shoulder letter; throughout the manuscript.
Occasionally, right shoulder letter “… more times6) → “… more times [6].;” “…. intervention)9” → “… intervention)[9]”; ….
In my opinion this manuscript requires revision before ready for publication in “Dianostics”.
Thank you for designating me as a peer reviewer.
Sincerely,
Susumu Matsukuma, MD, PhD,
Professor,
Department of Pathology and Laboratory Medicine,
National Defense Medical College,
3-2 Namiki, Tokorozawa City,
Saitama prefecture, 359-8513,
Japan.
E-mail: matsukuma@ndmc.ac.jp (College);
or skuma@cocoa.plala.or.jp (Home)
Author Response
Dear Dr. Matsukuma,
Thank you so much for your helpful comments and ideas. We shortened the manuscript and split Section 4 into two sections per your suggestion. We also removed the discussion section and shortened it into a conclusion per your feedback.
As far as the minor comments, we double checked all the reference numbers to ensure consistency and accuracy throughout.
Reviewer 2 Report
Section2:
I did not understand the comment regarding errors "those that are serious are usually promptly identified and corrected " If this occurs then what is the percentage that cause patient harm?
What percentage of radiology claims are due to communication breakdown?
Section 3:
I found the section on the Siewert study confusing. They reviewed their QA database and found communication errors in 421 out of 8701 patients, 4.8%. Of these they analysed 380 of which 144 (37.9%) had an effect on patient management . I am unsure where "85% of these errors being considered major" comes from as they report 89 of these were considered major.
Section 4:
I think it would be better to split the breast radiology out as a separate section to the proposed solution.
You mention all the issues which will negatively impact upon the introduction of the proposed solution and one potential strategy which when tested had significant impact on the communications - why did you choose not to include it in your model?
Section 6:
"Implementation of these ideals is only possible through legislation" whereas in section 4 " the process will be incentivised by the involvement of the covering insurance providers"
Both seem long term strategies - is there no other way to get buy in from the radiology departments?
Author Response
Thank you so much for your helpful comments and suggestions.
Section 2: We removed the sentence that you found unclear and also incorporated a statistic on the percentage of radiology claims due to communication breakdown per your suggestion.
Section 3: We reviewed the Siewert study and modified those sentences to make it more clear. The data we were trying to extract was this sentence from their paper: "In our study, 37.9% of communication errors had an impact on patient care, of which 85.4% were moderate or major events." We sincerely hope that we made it more clear now with our revisions. Thank you for bringing this to our attention.
Section 4: We split the breast radiology section out per your suggestion. We also included artificial intelligence within our schematic per your feedback as well.
Section 6: We removed that sentence regarding legislation and instead encouraged conversation with various policymakers to try and create incentives.
Round 2
Reviewer 1 Report
I think that the manuscript has been improved according to my comments.
Author Response
Thank you so much for your helpful comments and support!
Reviewer 2 Report
Thank you for your comments/changes.
I still find the sentence regarding the Siewert study confusing "These authors also found that communication errors affected 37.9% of patients, with over 85% of these errors being considered having a major impact on patient care (i.e., unnecessary imaging or surgical intervention)".
I can still read this as communication errors affected 37.9% of all radiology patients rather than .... of those patients who had communication errors 37.9% impacted on patient care with over 85% having a moderate or major impact (i.e., unnecessary imaging or surgical intervention)
Author Response
Thank you so much for your helpful comments. We modified the sentence per your feedback.
This manuscript is a resubmission of an earlier submission. The following is a list of the peer review reports and author responses from that submission.